# Contextual Hospital Conditions and the Risk of Nosocomial SARS-CoV-2 Infection: A Matched Case-Control Study with Density Sampling in a Large Portuguese Hospital

**DOI:** 10.3390/jcm13175251

**Published:** 2024-09-05

**Authors:** Francisco Almeida, Sofia Correia, Cátia Leal, Mariana Guedes, Raquel Duro, Paulo Andrade, Afonso Pedrosa, Nuno Rocha-Pereira, Carlos Lima-Alves, Ana Azevedo

**Affiliations:** 1Unidade de Prevenção e Controlo de Infeção e Resistência aos Antimicrobianos, Centro de Epidemiologia Hospitalar, Centro Hospitalar de São João, 4200-319 Porto, Portugal; francisco.almeida@ulssjoao.min-saude.pt (F.A.); mariana.guedes@ulssjoao.min-saude.pt (M.G.); paulo.andrade@chsj.min-saude.pt (P.A.);; 2Departamento de Ciências da Saúde Pública e Forenses e Educação Médica, Faculdade de Medicina da Universidade do Porto, 4200-319 Porto, Portugal; a.oliveira@chsj.min-saude.pt; 3EPIUnit, Instituto de Saúde Pública da Universidade do Porto, 4200-319 Porto, Portugalcatialeal1995@gmail.com (C.L.); 4Laboratório Para a Investigação Integrativa e Translacional em Saúde Populacional (ITR), Universidade do Porto, 4200-319 Porto, Portugal; 5Infectious Diseases and Microbiology Division, Hospital Universitario Virgen Macarena, Department of Medicine, Biomedicine Institute of Sevilla (IBiS)/CSIC, University of Sevilla, 41004 Sevilla, Spain; 6Unidade Local do Programa de Controlo de Infeção e Resistência aos Antimicrobianos, Unidade Local de Saúde do Tâmega e Sousa, 4560-136 Penafiel, Portugal; 7Serviço de Doenças Infeciosas, Unidade Local de Saúde do Tâmega e Sousa, 4560-136 Penafiel, Portugal; 8Serviço de Doenças Infeciosas, Centro Hospitalar de São João, 4200-319 Porto, Portugal; 9Serviço de Inteligência de Dados, Centro Hospitalar Universitário São João, 4200-319 Porto, Portugal; jose.pedrosa@chsj.min-saude.pt; 10Departamento de Medicina, Faculdade de Medicina da Universidade do Porto, 4200-319 Porto, Portugal; 11INFARMED—Autoridade Nacional do Medicamento e Produtos de Saúde, I.P., 1749-004 Lisboa, Portugal; 12Centro de Epidemiologia Hospitalar, Centro Hospitalar Universitário São João, 4200-319 Porto, Portugal

**Keywords:** SARS-CoV-2, respiratory transmission, transmission risk factors

## Abstract

**Objective:** Knowledge of the role of hospital conditions in SARS-CoV-2 transmission should inform strategies for the prevention of nosocomial spread of this pathogen and of similarly transmitted viruses. This study aimed to identify risk factors for nosocomial acquisition of SARS-CoV-2. **Methods:** We ran a nested case-control study with incidence density sampling among adult patients hospitalized for >7 days (August–December 2020). Patients testing positive for SARS-CoV-2 after the 7th day of hospitalization were defined as cases and matched with controls (1:4) by date of admission, hospitalization duration until index date, and type of department. Individual and contextual characteristics were gathered, including admission characteristics and exposures during the risk period. Conditional logistic regression was used to estimate the odds ratios (ORs) with respective 95% confidence intervals (CI) separately for probable (diagnosed on day 8–13) and definitive (diagnosed after day 14) nosocomial sets. **Results:** We identified 65 cases (31 probable; 34 definitive) and 219 controls. No individual characteristic was related to nosocomial acquisition of SARS-CoV-2. Contextual risk factors for nosocomial acquisition were staying in a non-refurbished room (probable nosocomial: OR = 3.6, 1.18–10.87), contact with roommates with newly diagnosed SARS-CoV-2 (probable nosocomial: OR = 9.9, 2.11–46.55; definitive nosocomial: OR = 3.4, 1.09–10.30), and contact with roommates with a first positive test 21–90 days before the beginning of contact (probable nosocomial: OR = 10.7, 1.97–57.7). **Conclusions:** Hospital conditions and contact with recently infected patients modulated nosocomial SARS-CoV-2 transmission. These results alert us to the importance of the physical context and of agile screening procedures to shorten contact with patients with recent infection.

## 1. Introduction

Nosocomial transmission of SARS-CoV-2 has been reported widely [1,2,3,4,5] and can pose both a risk to the individual patient and an increased burden on the functioning of healthcare services [6]. Studies performed in 2020 and 2021 have estimated the secondary attack rate of infection by the ongoing SARS-CoV-2 variants to be between 15 and 23% [7,8]. Populations of hospitalized patients often include frail individuals who remain at risk for severe complications from infection despite vaccination [9].

To prevent and mitigate nosocomial transmission, several non-pharmaceutical measures were adopted aiming towards “zero nosocomial infection” in various hospitals, particularly in those with centralized response teams and robust surveillance systems [10,11,12,13].

In addition to the epidemiological investigation attempting to mitigate an already established outbreak, it is fundamental to identify individual and contextual risk factors that can inform us of which policies and structural changes need to be implemented to decrease nosocomial spread of SARS-CoV-2 or similarly transmitted viruses.

This study aims to identify risk factors for nosocomial acquisition of SARS-CoV-2 in a Portuguese university hospital between August 2020 and December 2020 through a matched case-control study design.

## 2. Materials and Methods

Centro Hospitalar Universitário São João is a tertiary university public hospital in Porto, Portugal, with approximately 1100 beds. Construction of the main hospital building, where the study takes place, was finished in 1959. Many wards in the hospital are still organized according to the original hospital architecture and include rooms with up to 10 beds without private toilets. Ward refurbishment works have been in place for several years, which decreased the number of beds to up to three per room; implemented heating, ventilation, and air conditioning (HVAC) systems; and provided each room with a toilet. 

The study took place during the second wave of the SARS-CoV-2 pandemic in Portugal, when up to 10% of the study hospital’s beds were occupied with SARS-CoV-2-positive patients. During the study period, several protective measures were in place in the hospital, including universal masking for healthcare professionals and patients, visitor restrictions, postponement of all non-urgent surgical procedures, and changing of several workflows. Intensive training for healthcare professionals was provided, and guidelines for high-risk procedures were promptly adopted. The hospital-wide electronic surveillance system was adapted to guarantee timely and concerted action towards prevention of transmission. In the emergency room, patients with fever or respiratory symptoms were allocated in dedicated respiratory areas. 

Universal nasopharyngeal PCR screening for SARS-CoV-2 prior to hospital admission had been in place since March 2020. Regular screenings during hospitalization were progressively introduced in November 2020 (screening on the 5th day of hospitalization) and December 2020 (screening every 5 days of hospitalization). 

Patients who tested positive for SARS-CoV-2 were allocated to dedicated cohorts since the beginning of the pandemic. Patients who improved clinically were discharged from isolation after 20 days since symptom onset without the need for additional tests. For asymptomatic or mild infections in non-immunocompromised patients, there was the option of early discharge from isolation after 10 days if a negative nasopharyngeal swab PCR was obtained at that time.

Case-control eligibility criteria

We performed a nested case-control study with incidence density sampling. Adult individuals hospitalized between 1 August and 23 December 2020 and hospitalized for more than 7 days were eligible. Psychiatry, obstetrics, burn unit, and long-term hospitalization departments were excluded due to their particular characteristics and the low frequency of cases. Patients with a positive SARS-CoV-2 test or COVID-19 infection diagnosis before admission, at admission, or within 7 days from admission were also excluded.

Cases, controls, and matching

Cases were defined as patients with a first positive SARS-CoV-2 test after eight or more days of hospitalization, regardless of symptomatology (all patients had tested negative on admission). Cases were classified as either probable nosocomial (first positive test between 8 and 13 days) or definitive nosocomial (14 or more days after admission). Date of the first positive test was defined as the index date for each case. 

Each case was matched to a sample of patients who were still at risk of infection at the index date, i.e., with no positive test before and up to 14 days after index date (to exclude patients who could be in the incubation period) (Appendix A). Controls were selected randomly, with replacement, at a ratio of up to 4 controls per case and taking into account the matching variables. Matching was based on date of admission (+/−3 days), hospitalization day at index date (+/−1 day), and type of department at index date (medical, surgical, or intensive care). 

Risk factors

Several individual characteristics (inherent to the patient) and contextual factors (regarding hospital conditions) were gathered from the electronic health records. These factors were either non-variable characteristics (mainly related to the patient) or conditions that might change during hospitalization (mainly contextual aspects). For the latter, we considered conditions to which patients were exposed in the risk time window, i.e., in the 14 days before the index date (Figure 1). 

Individual characteristics included sex, age, comorbidities at admission (hypertension, chronic lung disease, heart failure, ischemic heart disease, diabetes mellitus, active neoplasm, transplant, and/or renal replacement therapy), and the highest level of patients’ dependency on hygiene, eating, or moving activities, recorded in the 14 days prior to index date.

To address hospital characteristics, we analyzed type of admission (urgent vs. elective), department at admission and index dates (further classified as medical, surgical, and intensive/intermediate care), emergency room (ER) stay, surgery, type of ward, in-hospital patient movement, and exposure to high-risk procedures and to other patients. These characteristics were assessed for the 14-day risk period. 

Patients’ movement within the hospital was defined as transition between departments and between rooms (number of different rooms up to index date). Rooms were further characterized as shared (≥1 bed) or private and as not refurbished (up to 10 beds per room, no in-room toilets) or refurbished (up to 3 beds per room, in-room toilets, heating, ventilation, and HVAC systems). Patients were classified as having stayed in a shared and in a non-refurbished room if they had been allocated at least once to one such room during their hospital stay. Additionally, the maximum number of beds in the rooms where the patient stayed was recorded. 

A roommate was defined as a patient who shared the same room as the case/control for any duration and regardless of the physical distance between assigned beds. 

Positivity for SARS-CoV-2 among roommates was recorded, which could have happened in the 90 days prior to the beginning of contact (SARS-CoV-2-positive roommates already discharged from isolation) or between the beginning of contact and the index date (newly diagnosed SARS-CoV-2 roommates). The total duration of contact with roommates and with SARS-CoV-2-positive roommates (in hours) was quantified.

Contact with a roommate exposed to high-risk aerosol-generating procedures in the 14 days prior to the index date was also recorded (detailed definition listed in Appendix A). 

Data were retrieved from the electronic information system through the in-house business intelligence platform HVITAL: the administrative software was SONHO v2, the laboratory software was Clinidata v5.3.1, the nursing registries and the medical diaries used SClinico v2. Some data were manually reviewed and screened for incongruences, as was the case with the dates of SARS-CoV-2 diagnosis and discharge from isolation. 

Statistical analysis

Cases and controls were characterized using descriptive statistics and stratified according to the type of nosocomial SARS-CoV-2 infection (probable vs. definitive). Categorical data were presented using frequency counts and percentages, and continuous data were presented using median with first and third quartiles. Conditional logistic regression was used to estimate crude and adjusted odds ratios (ORs) with respective 95% confidence intervals (CIs) for the entire sample and separately for probable and definitive nosocomial sets. We present the results for the crude and adjusted models. First set of adjustment was conducted for matching variables (admission and index dates and department at index date), and the final adjustment performed also included stay in non-refurbished rooms, which was significantly related with nosocomial infection in the crude analysis and could be a confounder of the evaluated exposures. 

Sensitivity analyses were performed to further understand possible risk factors: (a) restricting for those that were in non-refurbished rooms; (b) excluding patients who were in intensive care in the 14 days previously or at the index date. 

Analyses were performed with R software version 4.0.3.3. 

## 3. Results

After excluding nine patients without available controls for pairing, we included 65 cases matched with 219 controls selected from a population of 3089 eligible patients (Figure 2). Most cases were matched with four controls (59%), and 12% of cases were matched with one control. Around 25% of patients were controls of more than one case (most of them of two cases) and three definitive cases were controls from other cases. As expected, due to the longer hospital length of stay, a lack of controls was more likely to occur for definitive nosocomial cases.

Table 1 presents the distribution of the matching variables in cases and controls. The highest frequency of cases (22%) was observed between 23 and 29 November (week 48), followed by 11 cases two weeks earlier and 10 cases three weeks earlier. The median duration of hospitalization was 14 days for cases and controls; 10 days (interquartile range 8 to 10 days) for probable nosocomial cases and 23 days (19–30) for definitive nosocomial cases. Almost 70% of cases occurred in medical departments, particularly in Internal Medicine, and 25% in surgical departments, most frequently in Urology, General Surgery, and Neurosurgery (34% of definitive nosocomial cases and 17% of probable nosocomial cases). 

Table 2 presents the frequency of different characteristics among cases and controls in probable nosocomial and definitive nosocomial sets. Most patients were male and over 64 years of age. The majority of admissions had been urgent (non-elective) and to medical departments. Around 80% of cases and controls had one of the studied comorbidities, mostly hypertension, diabetes, and heart failure. 

In the 14-day risk period, 90% of probable nosocomial cases and 72% of controls had been to the emergency room. One fifth of the cases had undergone surgery (10% of probable and 30% of definitive nosocomial cases), as well as 25% of the controls. Around half of the controls and 68% of the cases were in a non-refurbished room. All the cases and 94% of the controls stayed in shared rooms. Patient transfer between rooms was frequent and occurred in 44% and 40% of the cases and controls. Around 30% of the cases were in contact with SARS-CoV-2-positive patients that were already discharged from isolation (vs. 18% of the controls). Almost 1/3 of the cases were in contact a patient who had a newly diagnosed SARS-CoV-2 infection during contact (vs. 11% of the controls). 

Table 3 presents the characteristics significantly associated with risk of acquisition of SARS-CoV-2 infection at the hospital, taking into account the confounding effect of type of department, date of admission, length of stay (model 2), and stay in non-refurbished wards (model 3). A full analysis, including all individual and contextual characteristics analyzed, can be found in Appendix A. We observed that no individual characteristic was related with nosocomial acquisition of SARS-CoV-2. Regarding contextual variables, having shared a room with patients who were newly diagnosed with SARS-CoV-2 infection during contact was the only risk factor for nosocomial infection, with association being found both among probable and definitive nosocomial cases. This association was evident even after adjusting for matching variables and potential confounders (OR 3.84 crude and 3.35 adjusted analysis for probable nosocomial cases, and OR 10.17 crude and 9.92 adjusted analysis for definitive nosocomial cases).

Additionally, for the probable nosocomial group, having stayed in a non-refurbished room at least once was associated with the acquisition of infection in the crude (OR 4.16) and adjusted analysis (OR 3.58). This association was not found in the definitive nosocomial group. 

Having shared a room with SARS-CoV-2-positive patients already discharged from isolation and having visited the emergency room in the 14 days prior to the index date were positively associated with acquisition of SARS-CoV-2 infection in the probable nosocomial group in the crude model, but these associations lost significance after adjustment.

Restricting the analysis to patients who had been admitted to non-refurbished wards showed no positive association with risk factors (Appendix A). In the analysis which excluded patients with a stay in intensive care in the 14 days prior to the positive test, acquisition of nosocomial SARS-CoV-2 was associated with the same risk factors found in the crude analysis of the whole dataset: stay in a non-refurbished room, sharing a room with a newly diagnosed SARS-CoV-2-infected patient, and sharing a room with a SARS-CoV-2-infected patient who had been discharged from isolation (Appendix A). 

## 4. Discussion

This study strengthens the notion that intra-hospital transmission is dependent mostly on contextual characteristics as opposed to individual vulnerability. 

The only factor associated both with probable and definitive nosocomial infection was sharing a room with newly diagnosed SARS-CoV-2 patients. This has been recognized as a major driver of nosocomial transmission of SARS-CoV-2. Risk of positivity in roommates of patients with newly diagnosed SARS-CoV-2 infections was estimated to be 40% in an institution with single or double rooms [14]. Hospital transmission occurs in clusters, and nosocomial outbreaks may have higher Rt than what is observed in community transmission [6].

Early recognition and isolation of SARS-CoV-2 patients is therefore a key step in preventing nosocomial transmission. At the height of the pandemic, widespread strategies included screening on admission, frequent testing during hospitalization, and contact tracing, but their efficacy can be hindered by the risk of transmission from asymptomatic or paucisymptomatic patients, which can last for several days [15,16]. Healthcare workers can also be a source of outbreaks or links in nosocomial chains of transmission [6]. Since healthcare workers were not routinely tested in our institution, we cannot estimate their putative contribution. During periods of elevated community transmission, universal testing is a valid complement to daily symptom screening as strategies to hasten identification and isolation of patients with acute SARS-CoV-2 infection, especially if broad criteria are used and if asymptomatic patients are included [6,17].

Room conditions were associated with increased risk but only among probable nosocomial cases. For other types of nosocomial respiratory infections, rooms’ occupancy and bed distance, supporting ventilation systems, and type of materials/equipment have been recognized as important drivers of transmission [5]. In particular, improved ventilation has been postulated as an important measure in curbing indoor transmission of SARS-CoV-2, including in the hospital setting [18]. Reducing the number of patients per room lowers the number of patients exposed to any given SARS-CoV-2 case. Also, increasing the availability of single-patient rooms has been recognized as an effective measure in the prevention of nosocomial transmission of infections, including SARS-CoV-2 [19,20]. Surprisingly, the risk of transmission was not specifically associated with the number of beds per room in our study, which may point to the importance of the role played by air conditioning with high-efficiency particulate air filters and non-shared bathrooms in reducing transmission in the refurbished rooms.

We observed an unexpected association between the risk of nosocomial acquisition of SARS-CoV-2 and contact with patients with prior SARS-CoV-2 infection already discharged from isolation. This association was only found in the probable nosocomial infection group (where it lost significance after adjustment) and in the sensitivity analysis, where patients who had exposure to intensive care during the risk period were excluded.

The duration of infectivity in SARS-CoV-2 infection is disputed: viable virus shedding, a surrogate marker of infectivity, is thought not to surpass 10 days in non-severe cases in immunocompetent patients [21] and 20 days in severe or critical cases [22] for pre-omicron strains. Some agencies such as the ECDC have recommended antigen testing before discharge from isolation [23], which was not routinely performed during the study period in our institution. For severely immunosuppressed patients, a negative molecular test was required for terminating isolation precautions, along with a 14-day period after symptom onset and clinical improvement. However, cases of viable virus shedding persisting for months have been reported [24,25,26] in this patient population. Therefore, it is possible that some of the roommates already discharged from SARS-CoV-2 isolation were still shedding viable virus and posed a transmission risk.

The main strength of our study was the possibility to include in the analysis contextual factors which are cumbersome to obtain through manual review of electronic health records. However, errors in automatic processes of data extraction, such as those related to recording, may be difficult to correct. Manual review was performed in order to correct incongruent data, but we cannot exclude the possibility that some errors might have persisted and resulted in misclassification of the evaluated characteristics.

This case-control study used a density incident sampling process, which allowed us to select a representative sample of patients at risk within the cohort of interest, minimizing bias related to its retrospective nature. Additionally, several contextual factors change during hospital stay, and it is often impossible to retrieve from databases the exact hospitalization moment when an exposure occurred. This study was able to define the exposure in the 14 days before the index date, decreasing reverse causation and improving causal reasoning.

Our study has several limitations. Firstly, this study was not an outbreak investigation and, thus, did not intend to evaluate transmission as an individual event for each patient. However, staying in a room or ward where an outbreak is ongoing would likely be an independent risk factor for SARS-CoV-2 acquisition, as described in a recent paper by Aghdassi et al. [27]. Also, outbreaks were not labeled as such in our database, and therefore, we were unable to differentiate between exposure to one newly diagnosed SARS-CoV-2 patient and exposure to ongoing transmission with several cases. Additionally, our study was probably underpowered to detect relevant associations, as shown by the wide confidence intervals in our outcome measures. Moreover, some relevant comorbidities such as obesity were not systematically recorded in the electronic records and, therefore, not studied as potential risk factors. Furthermore, healthcare workers were not tested routinely, and we were not able to assess which patients we in contact with healthcare workers diagnosed with SARS-CoV-2 infection, since it is also not possible to retrieve patient–healthcare worker contact information from our electronic records. Similarly, we were unable to study the role of visitors in transmission, but this mechanism likely played a minor role during the study period, since visits to the hospital were restricted according to national policy. In addition, specific hospital areas such as long-term care departments were excluded from the analysis due to their different characteristics and low incidence of nosocomial SARS-CoV-2 during the study period. Nosocomial SARS-CoV-2 transmission in those settings might be driven by different risk factors which should be assessed in further studies. Finally, the behavior of the pandemic during the study period is likely divergent from what would be currently found, or from what would be found in the future, due to different circulating SARS-CoV-2 strains and the onset of vaccination. However, the incidence of nosocomial cases has been reported to remain high even with more-recent strains and in vaccinated populations [28], particularly after discontinuation of infection control measures [29,30]. In these contexts, despite significantly lower severity, SARS-CoV-2 mortality is still relevant in certain patient populations [28], and therefore, it remains relevant to understand which features of the hospital setting are driving transmission.

In conclusion, early identification of SARS-CoV-2-positive patients is likely the main tool in mitigating nosocomial transmission. Testing policy should be informed by local epidemiology, and an increase in the number of cases should be met with routine frequent and transversal testing within the hospital. Also, our results are consistent with the literature that suggests that building or renovating hospital structures according to the principles of infection prevention will play a role in mitigating transmission of respiratory viruses.

## Figures and Tables

**Figure 1 jcm-13-05251-f001:**
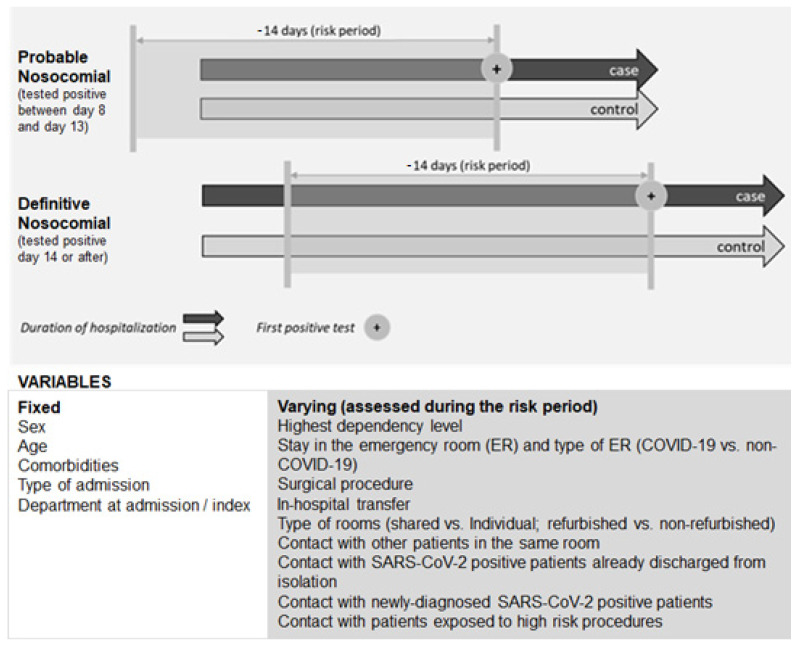
Risk factors evaluated and risk period considered.

**Figure 2 jcm-13-05251-f002:**
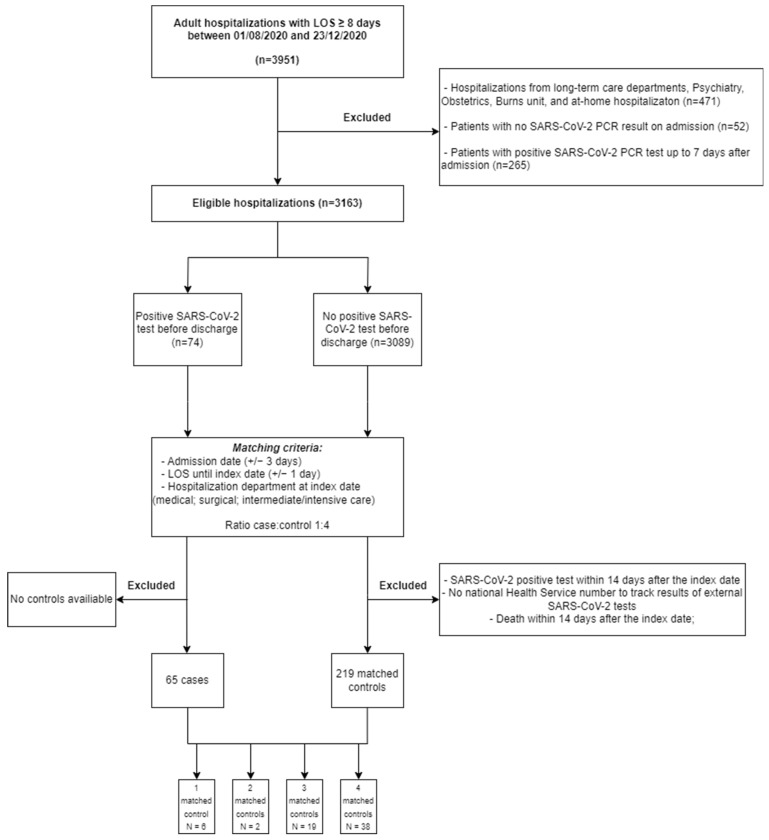
Flowchart for cases and controls selection.

**Table 1 jcm-13-05251-t001:** Matching criteria.

	Nosocomial SARS-CoV-2 Cases (n = 65)	Controls (n = 219)
Week of the admission date ^a,b^	44 (41; 46)	44 (41; 46)
Length of stay until index date ^a^	14.0 (10.0; 23.4)	14.0 (9.0; 21.0)
Type of department at index date % (n)		
Surgical	17 (25.2)	56 (25.6)
Medical	45 (69.2)	156 (71.2)
Intensive care	3 (4.6)	7 (3.2)

^a^ Median (percentile 25; percentile 75); ^b^ week of the year (within 52 weeks; minimum = 32 (3–9 August 2020); maximum = 50 (7–13 December 2020)).

**Table 2 jcm-13-05251-t002:** Characteristics of cases and controls.

	SARS-CoV-2 Nosocomial Transmission			
	Probable		Definitive	
	Controls	Cases	Controls	Cases
	(n = 114)	(n = 31)	(n = 105)	(n = 34)
Age (years)				
18–54	30 (26.3)	6 (19.4)	29 (27.6)	7 (20.6)
55–64	17 (14.9)	6 (19.4)	20 (19.0)	4 (11.8)
65–74	27 (23.7)	5 (16.1)	30 (28.6)	11 (32.4)
≥75	40 (35.1)	14 (45.2)	26 (24.8)	12 (35.3)
Male	63 (55.3)	17 (54.8)	64 (61.0)	19 (55.9)
Urgent admission	87 (76.3)	28 (90.3)	80 (76.2)	29 (85.3)
Type of admission department				
Surgical	21 (18.4)	5 (16.1)	23 (21.9)	10 (29.4)
Medical	74 (64.9)	19 (61.3)	53 (50.5)	20 (58.8)
Intensive care	19 (16.7)	7 (22.6)	29 (27.6)	4 (11.8)
Dependent patient	78 (68.4)	24 (77.4)	45 (42.9)	13 (38.2)
Comorbidities	95 (83.3)	22 (71.0)	84 (80.0)	29 (85.3)
Arterial hypertension	57 (50.0)	15 (48.4)	53 (50.5)	19 (55.9)
Chronic obstructive pulmonary disease	15 (13.2)	6 (19.4)	9 (8.6)	3 (8.8)
Heart failure	30 (26.3)	6 (19.4)	13 (12.4)	9 (26.5)
Ischemic heart disease	19 (16.7)	5 (16.1)	14 (13.3)	4 (11.8)
Diabetes mellitus	21 (18.4)	8 (25.8)	34 (32.4)	14 (41.2)
Active neoplasm	34 (29.8)	5 (16.1)	29 (27.6)	6 (17.6)
Transplant	4 (3.5)	0 (0.0)	5 (4.8)	0 (0.0)
Renal replacement therapy	4 (3.5)	0 (0.0)	2 (1.9)	1 (2.9)
Contextual characteristics (14 days prior to index date)				
Emergency room visit ^a^	82 (71.9)	28 (90.3)		
Surgeries	23 (20.2)	3 (9.7)	31 (29.5)	10 (29.4)
Stay in a non-refurbished room	54 (47.4)	24 (77.4)	54 (51.4)	20 (58.8)
Number of different rooms				
1	71 (62.3)	15 (48.4)	59 (56.2)	21 (61.8)
2	34 (29.8)	11 (35.5)	26 (24.8)	9 (26.5)
3	9 (7.9)	5 (16.1)	20 (19.0)	4 (11.8)
Stay in a shared ward	106 (93.0)	31 (100.0)	100 (95.2)	34 (100.0)
Maximum number of beds in room				
1	8 (7.0)	0 (0.0)	5 (4.8)	0 (0.0)
2–4	39 (34.2)	8 (25.8)	42 (40)	16 (47.1)
5–9	51 (44.7)	17 (54.8)	32 (30.5)	15 (44.1)
≥10	16 (14)	6 (19.4)	26 (24.8)	3 (8.8)
Contact with other patients	106 (93.0)	31 (100.0)	100 (95.2)	33 (97.1)
Duration of contact (hours)				
≤750	60 (56.6)	16 (51.6)	39 (39.0)	13 (39.4)
751–1500	34 (32.1)	12 (38.7)	33 (33.0)	14 (42.4)
>1500	12 (11.3)	3 (9.7)	28 (28.0)	6 (18.2)
Contact with patients exposed to high-risk procedures	64 (56.1)	21 (67.7)	51 (48.6)	18 (52.9)
Contact with SARS-CoV-2-positive patients already discharged from isolation	30 (26.3)	14 (45.2)	9 (8.6)	6 (17.6)
Contact with newly diagnosed SARS-CoV-2-positive patients	12 (10.5)	9 (29.0)	12 (11.4)	12 (35.3)

^a^ Only for probable cases and respective controls.

**Table 3 jcm-13-05251-t003:** Characteristics associated with risk for nosocomial SARS-CoV-2 acquisition.

	Probable (n = 145)			Definitive (n = 139)		
	Crude Analysis	M2 ^a^	M3 ^b^	Crude Analysis	M2 ^a^	M3 ^b^
Stay in non-refurbished wards	4.16 (1.59–10.85)	4.78 (1.65–13.83)	3.58 (1.18–10.87)	1.27 (0.57–2.81)	1.14 (0.49–2.66)	0.72 (0.27–1.91)
Sharing room with SARS-CoV-2-positive patients already discharged from isolation	2.73 (1–7.48)	3.2 (1.03–9.92)	2.51 (0.78–8.03)	2.44 (0.75–7.94)	2.23 (0.68–7.38)	2.27 (0.65–7.93)
Sharing room with newly diagnosed SARS-CoV-2-positive patients	3.84 (1.37–10.72)	3.85 (1.35–11.02)	3.35 (1.09–10.3)	10.17 (2.2–46.97)	9.91 (2.13–46.2)	9.92 (2.11–46.55)

^a^ After adjusting for matching variables; ^b^ after adjusting for matching variables and stay in non-refurbished wards.

## Data Availability

The data presented in this study are available on request from the corresponding author due to ethical reasons.

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
