# Peer review of "Contextual Hospital Conditions and the Risk of Nosocomial SARS-CoV-2 Infection: A Matched Case-Control Study with Density Sampling in a Large Portuguese Hospital"

_jcm, 2024, doi:10.3390/jcm13175251_

Round 1
Reviewer 1 Report
Comments and Suggestions for Authors
Some observations for clarification and/or correction for the authors:
In lines 45, 46, 154, 155, 156, reference is made to the consideration of contact with roommates with a positive PCR even up to 90 days prior to the index date of the cases as a risk factor. Indeed, in the literature there are reports of viable virus even for months in respiratory secretions of immunosuppressed patients, (in fact, I think it is good that this is pointed out in lines 293 and 294 of the discussion section) however, in the present study there is no reference to this prolonged risk period being considered for roommates in this immunological condition. Could you clarify this?
I believe that lines 72 and 73 should be deleted.
Could you explain why, if hospital stay in remodeled wards seemed to be a "protective factor" for nosocomial SARS-CoV-2 infection, this was not reflected in the stay in wards with fewer beds compared to stays in wards with 10 shared beds, for example? Was it more important to have air conditioning and non-shared bathrooms? Did air conditioners include high-efficiency filters with sanitary-hospital characteristics?
Is it not clear why patients from long-term care departments were excluded from the study? because in these areas, patients with risk co-morbidities for COVID-19 are frequently found, such as diabetes, severe obesity, cardiovascular disease and chronic lung disease, for example. (lines 103, 104 and 105). Please clarify.
At the top of Figure 1, it is probably more illustrative if it is noted that probable nosocomial patients had a positive test between day eight and 13 of hospitalization and definitive nosocomial patients had a positive test after 14 days of hospitalization.
In many studies on COVID-19, obesity has been identified as a risk factor for poor prognosis, even lethality. Why was this factor not considered in this study?
Indeed, as the authors point out in lines 267 and 268, SARS CoV-2 infection in health workers can represent a risk even for nosocomial outbreaks. The authors point out that this risk factor was not analyzed because these workers were not regularly and systematically screened; However, a brief analysis comparing contact with health workers with COVID-19 during the risk period between cases and controls, or at least a description of the cases in which this scenario existed, may be included.
Author Response
In lines 45, 46, 154, 155, 156, reference is made to the consideration of contact with roommates with a positive PCR even up to 90 days prior to the index date of the cases as a risk factor. Indeed, in the literature there are reports of viable virus even for months in respiratory secretions of immunosuppressed patients, (in fact, I think it is good that this is pointed out in lines 293 and 294 of the discussion section) however, in the present study there is no reference to this prolonged risk period being considered for roommates in this immunological condition. Could you clarify this? -
Thank you for your remark. During the study period, isolation precautions for severely immunosuppressed SARS-CoV-2 patients were suspended after the 14th day since de beginning of symptoms if clinical improvement and a negative SARS-CoV-2 molecular test was present. Despite this, we agree that some of these patients might have had the potential to transmit the disease after discharge from isolation. We have added the description of our criteria to suspend isolation in immunosuppressed patients to the manuscript
I believe that lines 72 and 73 should be deleted.
Than you for pointing out this mistake. We have removed these lines
Could you explain why, if hospital stay in remodeled wards seemed to be a "protective factor" for nosocomial SARS-CoV-2 infection, this was not reflected in the stay in wards with fewer beds compared to stays in wards with 10 shared beds, for example? Was it more important to have air conditioning and non-shared bathrooms? Did air conditioners include high-efficiency filters with sanitary-hospital characteristics?
We agree and were also expecting to find higher risk among patients in large wards vs wards with fewer beds. We believe that either our study was underpowered do find an association here (which we have already stated as a limitation) or that the differentiating factor in refurbished rooms were the presence of non-shared bathrooms and air conditioning with HEPA filters, as you pointed out. We have elaborated more on those conclusions in the text.
Is it not clear why patients from long-term care departments were excluded from the study? because in these areas, patients with risk co-morbidities for COVID-19 are frequently found, such as diabetes, severe obesity, cardiovascular disease and chronic lung disease, for example. (lines 103, 104 and 105). Please clarify.
This point is well taken. We opted to not include long term care rooms because) the risk of transmission is different from acute care wards and is probably driven by different characteristics, such as the individual factors suggested in the reviewer’s query. Including long-term care departments would probably add more differentiated information but it would make our results more difficult to apply to the acute care setting. Also, at the time there were only a few nosocomial SARS-CoV-2 cases in our long-term care department, which is located in a separate hospital building and has a wholly separate staff. We have added this as a limitation of the article
At the top of Figure 1, it is probably more illustrative if it is noted that probable nosocomial patients had a positive test between day eight and 13 of hospitalization and definitive nosocomial patients had a positive test after 14 days of hospitalization.-
Thank you for your suggestion. We have edited Figure 1 accordingly
In many studies on COVID-19, obesity has been identified as a risk factor for poor prognosis, even lethality. Why was this factor not considered in this study?
We agree that obesity is a relevant comorbidity in the setting of SARS-CoV-2 infection. Unfortunately, patient weight is not systematically recorded in our electronic record, so such data was not possible to obtain automatically;. We have added this as a limitation to the text.
Indeed, as the authors point out in lines 267 and 268, SARS CoV-2 infection in health workers can represent a risk even for nosocomial outbreaks. The authors point out that this risk factor was not analyzed because these workers were not regularly and systematically screened; However, a brief analysis comparing contact with health workers with COVID-19 during the risk period between cases and controls, or at least a description of the cases in which this scenario existed, may be included.
We agree this information would be relevant not only for a better understanding of our data but also as a practical tool to break chains of transmission. Sadly, we were unable to access data on SARS-CoV-2 infection in healthcare workers in a way that could be correlated with geographic areas or with patients at risk. We have added this note to the manuscript
Reviewer 2 Report
Comments and Suggestions for Authors
This is a well performed study looking at nosocomial risk factors for transmission of COVID in a single hospital.
The manuscript is well written.
My main comment is that the authors present a large amount of data comparing COVID patients with controls, that is not statistically significant.
This blur the key message, and I suggest that non-significant results are deleted from table 1, 2 and 3, and just briefly mentioned in the results that the rest of the comparisons were not significant.
Author Response
My main comment is that the authors present a large amount of data comparing COVID patients with controls, that is not statistically significant.
This blur the key message, and I suggest that non-significant results are deleted from table 1, 2 and 3, and just briefly mentioned in the results that the rest of the comparisons were not significant.
Thank you for your suggestion. We agree and have simplified the table as suggested. The article became indeed easier to read and the message clearer. We have included the full table in the supplements.
Reviewer 3 Report
Comments and Suggestions for Authors
General comments:
The article is well designed and presented.
Specific comments:
Results:
In flowchart would it could be interesting to individualize the three reasons for exclusion from the study (ward of hospitalization, no PCR test performed, PCR positive during first 7 days). What is Valongo?
What is the percentage of patients with PCR positive in the hospital during the period analysed?
References:
There are no references for the current year and only two for 2023.
Author Response
In flowchart would it could be interesting to individualize the three reasons for exclusion from the study (ward of hospitalization, no PCR test performed, PCR positive during first 7 days).
Thank you for your suggestion. We have edited the flowchart accordingly
What is Valongo? We thank you for pointing out this mistake. “Valongo” is the local designation of the separate building of the hospital where long care wards are located. We have replaced this designation with “Patients admitted to long term care wards”
What is the percentage of patients with PCR positive in the hospital during the period analysed?
The percentage of patients admitted with SARS-CoV-2 varied widely during the study period, in consonance with the epidemiological curve of the pandemic in the north of Portugal in late 2020. At its peak approximately 10% of beds were occupied with SARS-CoV-2 patients. The number of nosocomial cases during that period was 74, as described in the flowchart. We have added some of this information to the part of the methods section where we describe the study setting
There are no references for the current year and only two for 2023.
We appreciate this input. We recognize that we should update the literature review, although many relevant studies were published between 2020 and 2022. We have added more recent references.
Round 2
Reviewer 2 Report
Comments and Suggestions for Authors
No further comments